# The neurocognitive role of working memory load when Pavlovian motivational control affects instrumental learning

**Heesun Park**[1☺], **Hoyoung Doh**[1☺], **Eunhwi Lee**[1], **Harhim Park**[1], **Woo-Young Ahn**[1,2]*

**1** Department of Psychology, Seoul National University, Seoul, Korea, **2** Department of Brain and Cognitive Sciences, Seoul National University, Seoul, Korea

☺ These authors contributed equally to this work.
* wahn55@snu.ac.kr

**Data Availability Statement:** Behavioral data and codes for behavioral analysis and modeling are available at https://github.com/CCS-Lab/project_WMDM_public. Unthresholded t maps are

## Abstract

Research suggests that a fast, capacity-limited working memory (WM) system and a slow, incremental reinforcement learning (RL) system jointly contribute to instrumental learning. Thus, situations that strain WM resources alter instrumental learning: under WM loads, learning becomes slow and incremental, the reliance on computationally efficient learning increases, and action selection becomes more random. It is also suggested that Pavlovian learning influences people's behavior during instrumental learning by providing hard-wired instinctive responses including approach to reward predictors and avoidance of punishment predictors. However, it remains unknown how constraints on WM resources affect instrumental learning under Pavlovian influence. Thus, we conducted a functional magnetic resonance imaging (fMRI) study (N = 49) in which participants completed an instrumental learning task with Pavlovian–instrumental conflict (the orthogonalized go/no-go task) both with and without extra WM load. Behavioral and computational modeling analyses revealed that WM load reduced the learning rate and increased random choice, without affecting Pavlovian bias. Model-based fMRI analysis revealed that WM load strengthened RPE signaling in the striatum. Moreover, under WM load, the striatum showed weakened connectivity with the ventromedial and dorsolateral prefrontal cortex when computing reward expectations. These results suggest that the limitation of cognitive resources by WM load promotes slow and incremental learning through the weakened cooperation between WM and RL; such limitation also makes action selection more random, but it does not directly affect the balance between instrumental and Pavlovian systems.

## Author summary

Among multiple decision-making systems of humans and animals, the Pavlovian system is known for promoting automatic and instinctive behaviors. Understanding the Pavlovian influence on decision-making can offer valuable insights into the mechanism of impulsive and addictive behaviors. Previous studies suggested that prefrontal executive control can be important in regulating the Pavlovian influence. We tested if reducing

available on Neurovault: https://identifiers.org/neurovault.collection:12860. We have shared raw neuroimaging data in OpenNeuro: https://openneuro.org/datasets/ds004647/versions/1.0.2.

**Funding:** This work was supported by the Basic Science Research Program through the National Research Foundation (NRF) of Korea (https://nrf.re.kr/) funded by the Ministry of Science and ICT (MSIT) (https://www.msit.go.kr/) (NRF-2018R1C1B300731315 and NRF-2018R1A4A102589113), the National Research Foundation of Korea (NRF) grant funded by the Korea government (MSIT) (No. 2021M3E5D2A0102249311), the Creative-Pioneering Researchers Program through Seoul National University (https://www.snu.ac.kr/), and the BK21 FOUR (https://bk21four.nrf.re.kr/) program [5199990314123] to W.-Y.A. The funders had no role in study design, data collection and analysis, decision to publish, or preparation of the manuscript.

**Competing interests:** The authors declare no competing financial interests.

cognitive resources available for executive control modulates the Pavlovian influence, by adding WM load to an instrumental learning task where Pavlovian influence is beneficial in some conditions but detrimental in others. Contrary to our expectation, constraining cognitive resources with WM load failed to significantly change the contribution of Pavlovian system. Nonetheless, with behavioral analysis and computational modeling, we revealed that WM load promotes slower learning and makes decisions noisier in an instrumental learning task with Pavlovian components. FMRI analysis revealed that WM load strengthens the RPE signaling in striatum upon observing the outcome and weakens the functional connectivity between the prefrontal cortex and the striatum before making a decision. The current study contributes to understanding how cognitive resource constraints alter learning and decision-making under Pavlovian influence as well as the neural mechanisms of those effects.

## Introduction

The process of learning about the environment from experience and making adaptive decisions involves multiple neurocognitive systems, among which reinforcement learning (RL) and working memory (WM) systems are known to significantly contribute to learning [1–3]. RL processes facilitate "incremental" learning from the discrepancy between actual and predicted rewards, known as reward prediction error (RPE); RL is regarded as a slow but steady process [4]. Dopaminergic activity in the basal ganglia conveys RPEs [5–11], and human imaging studies have found that blood-oxygen-level-dependent (BOLD) signals in the striatum are correlated with RPEs [12–14].

In addition to RL, WM is a crucial component in learning. In particular, WM allows the rapid learning of actions via retention of recent stimulus-action-outcome associations, while RL constitutes a slow learning process [1,15–17]. WM can also offer various inputs to RL, such as reward expectations [18] and models of the environment [19–22] as well as complex states and actions [3,23]. In the brain, the WM system is presumably associated with sustained neural activity throughout the dorsolateral prefrontal cortex (dlPFC) and prefrontal cortex (PFC) [24–29].

Because RL and WM cooperate to promote successful learning, restricting either system can alter learning and the balance between the two systems. In particular, increasing WM load during learning and decision-making can lead to various consequences through the depletion of WM resources. For example, first, instrumental learning becomes slow and incremental under WM load [1,16,30,31]. Limited resources in the WM system cause WM contribution to decline while the RL contribution increases, causing learning to occur more slowly and strengthening the RPE signal in the brain [15,18]. Second, among the multiple RL systems that use varying degrees of WM resources, the demands of WM can be balanced against computationally costly strategies. Otto et al. demonstrated that under WM load, the reliance on computationally efficient model-free learning was increased, compared with model-based learning [32]. Lastly, limited WM resources may cause action selection to become more random and inconsistent. Different values must be compared to inform decision-making during the action selection stage [33], but several studies have reported that WM load may interrupt these processes without affecting valuation itself [34–36].

While reductions of WM resources substantially alter instrumental learning, another important factor known to shape instrumental learning is the Pavlovian system. Through the motivation of hard-wired responses, such as active responses to appetitive cues and inhibitory

responses toward aversive cues [37–40], the Pavlovian system may facilitate certain instrumental behaviors and impede others. This bias in instrumental learning is known as Pavlovian bias [41–44]. In a widely-used paradigm, Pavlovian bias emerges when the cues used in a instrumental learning task trigger Pavlovian conditioned responses (e.g., active responses to reward cues regardless of action requirements), leading to an increase or decrease in task performance [45,46]. Many computational models of the phenomenon assume that decision-makers consider both instrumentally trained responses and Pavlovian conditioned responses when deciding which response to give in each trial. This decision-making procedure balances the instrumental and Pavlovian learning systems, where the instrumental system is more flexible but computationally more expensive while the Pavlovian system is more rigid but computationally cheaper. This is because the instrumental system learns stimulus-response-outcome associations, whereas the Pavlovian system only learns stimulus-outcome associations and has hard-wired responses for the learned associations. Pavlovian bias is generally presumed to be associated with maladaptive behaviors such as substance use disorder and compulsivity-related disorders [47–51].

Although it is well known that increasing WM load alters instrumental learning in several ways, it remains unclear how WM load changes instrumental learning when it is under Pavlovian influence. To investigate this relationship, we conducted a functional magnetic resonance imaging (fMRI) study in which participants completed an instrumental learning task that involved Pavlovian–instrumental conflicts [45], with and without extra WM load.

We tested the following three hypotheses. First, if the role of WM in learning is unaffected by Pavlovian influence, WM load will lead to slower learning and increased striatal RPE signals, consistent with previous findings [1,15,16,18]. Second, if WM load leads to a computational trade-off between Pavlovian and instrumental learning, like the one between model-free and model-based learning [32], WM load will enhance Pavlovian bias because the Pavlovian system is known to require fewer resources and to be computationally efficient as an evolutionarily embedded system that learns values as a function of cues, regardless of actions [42]. We also presumed that neural signaling associated with Pavlovian bias would increase under WM load. We focused on regions of the basal ganglia, specifically the striatum and substantia nigra/ventral tegmental area (SN/VTA), which are considered important in Pavlovian bias [45,46,52–54]. Third, if the contribution of WM to consistent action selection remains consistent, WM load will cause action selection to become more random, as in previous studies [34–36]. We tested whether the value comparison signal in the brain would decrease under WM load because consistent action selection may be associated with the extent to which value difference information is utilized during the decision-making process [33,55].

Our behavioral and computational modeling results revealed that Pavlovian bias did not increase under WM load, while learning became slower and more incremental and action selection became more random. We observed increased striatal RPE signaling which was in line with our hypothesis that the contribution of RL will increase under WM load. Further analyses revealed weakened connectivity between the striatal and prefrontal regions under WM load, suggesting diminished cooperation between the WM and RL systems.

## Results

The participants (N = 56) underwent fMRI imaging while performing an instrumental learning task under a control condition and a WM load condition (**Fig 1**). In the control condition, they participated in the orthogonalized go/no-go (GNG) task [45], a learning task that contained Pavlovian–instrumental conflicts. In the WM load condition, a 2-back task was added to the GNG task; the modified task was named the working memory go/no-go (WMGNG) task (see Materials and Methods for more detail).

**A**

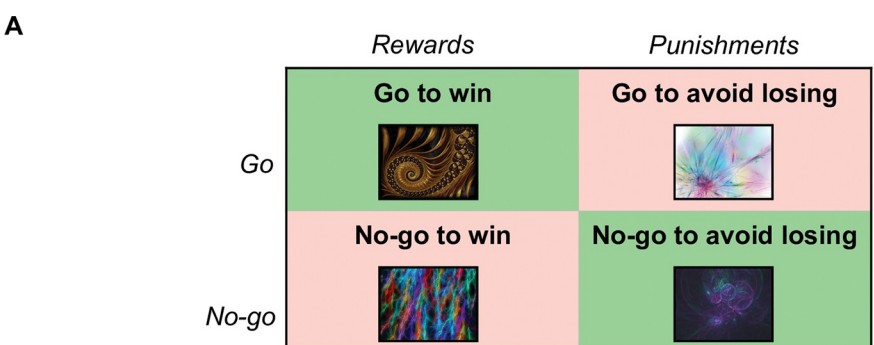

**B**

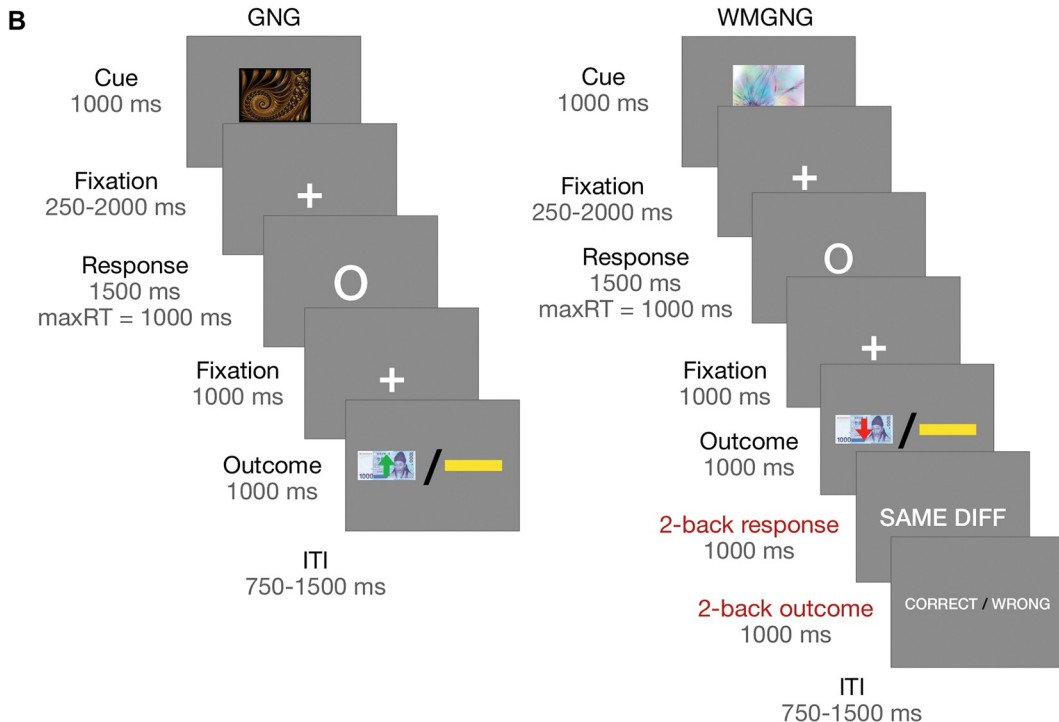

**C**

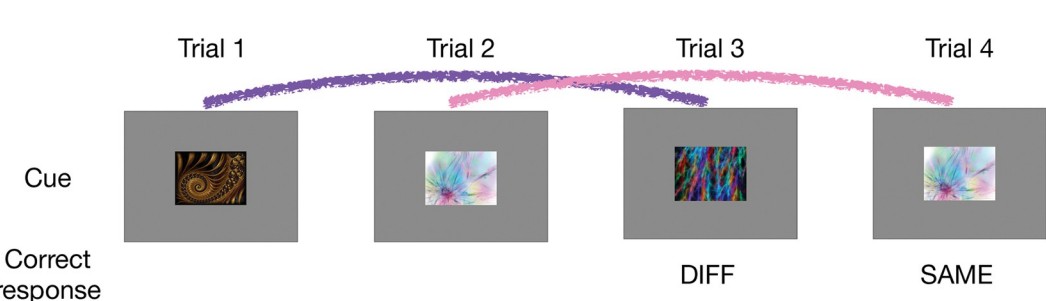

**Fig 1. The GNG and WMGNG tasks.** (A) In both tasks, four fractal cues indicated the combination of action (go/no-go) and valence at the outcome (win/loss). (B) In each trial, a fractal cue was presented, followed by a variable delay. After the delay, actions were required in response to a circle, and participants had to decide whether to press a button. After an additional brief delay, the probabilistic outcome was presented, indicating monetary reward (green upward arrow on a ₩1000 bill) or monetary punishment (red downward arrow on a ₩1000 bill). A yellow horizontal bar indicated no win or loss. In the

WMGNG task, the original GNG task was followed by a 2-back response and 2-back outcome phases. (C) The participants were asked to indicate whether the cue in the current trial was identical to the cue in the two preceding trials. Here, because the cue in trial 3 differed from the cue in trial 1, "DIFF" was the correct response. Similarly, because the cue in trial 4 was identical to the cue in trial 3, "SAME" was the correct response. The lines mark two cues for comparison: the purple line indicates that the cues differ, while the pink line indicates that the cues are identical.

## Task performance: Decreased performance and learning speed under WM load

Imposing extra WM load with a 2-back task led to a decrease in task accuracy. Participants performed better in the GNG task ($M = 0.80$, $SD = 0.12$) than in the WMGNG task ($M = 0.72$, $SD = 0.16$), as illustrated in **Fig 2A** (paired t-test, $t(48) = 3.86$, $p<0.001$, $d = 0.55$). Participants' performance decreased both in the Pavlovian-congruent ("go to win" and "no-go to avoid losing") and Pavlovian-incongruent ("no-go to win" and "go to avoid losing") conditions (**S1A Fig**).

We also confirmed that participants exhibited go bias and Pavlovian bias in both tasks, thus replicating the findings of earlier studies [45,52,56–62]. Two-way ANOVA on accuracy, with the factors action (go/no-go) and valence (reward/punishment) as repeated measures for both tasks, revealed a main effect of action ($F(48) = 6.05$, $p = 0.018$, $\eta^2 = 0.03$ in GNG task, $F(48) = 9.44$, $p = 0.003$, $\eta^2 = 0.04$ in WMGNG task) and action by valence interaction ($F(48) = 22.43$, $p<0.001$, $\eta^2 = 0.12$ in the GNG task, $F(48) = 30.59$, $p<0.001$, $\eta^2 = 0.10$ in the WMGNG task); it showed no effect of valence ($F(48) = 0.00$, $p = 0.99$, $\eta^2 = 0.00$ in the GNG task, $F(48) = 2.77$, $p = 0.103$, $\eta^2 = 0.01$ in the WMGNG task). In both tasks (**Fig 2B**), participants exhibited superior performances in "go to win" and "no-go to avoid losing" conditions (i.e., Pavlovian-congruent conditions; blue columns) than in "no-go to win" and "go to avoid losing" trials (i.e., Pavlovian-incongruent conditions; red columns). Specifically, in the GNG task, accuracy was higher in the "go to win" ($M = 0.92$, $SD = 0.12$) than "no-go to win" condition ($M = 0.69$, $SD = 0.35$) (paired t-test, $t(48) = 4.13$, $p<0.001$, $d = 0.59$), and in the "no-go to avoid losing" ($M = 0.85$, $SD = 0.13$) than in the "go to avoid losing" condition ($M = 0.76$, $SD = 0.18$) (paired t-test, $t(48) = 3.29$, $p = 0.002$, $d = 0.47$). Similarly, in the WMGNG task, accuracy was higher in the "go to win" ($M = 0.82$, $SD = 0.25$) than in the "no-go to win" condition ($M = 0.57$, $SD = 0.34$) (paired t-test, $t(48) = 4.82$, $p<0.001$, $d = 0.69$), and in the "no-go to avoid losing" ($M = 0.79$, $SD = 0.16$) than in the "go to avoid losing" condition ($M = 0.72$, $SD = 0.19$) (paired t-test, $t(48) = 2.51$, $p = 0.015$, $d = 0.36$).

Next, we tested the hypothesis that WM load would decrease learning speed (**Fig 2C**). While the learning curves indicated that participants learned during both tasks, the learning curve was shallower in the WMGNG task than in the GNG task (i.e., WM load reduced learning speed and overall accuracy).

To test the hypothesis that WM load would increase Pavlovian bias (**Fig 2D**), we quantified Pavlovian bias by subtracting the accuracy in Pavlovian-incongruent conditions ("no-go to win" and "go to avoid losing") from the accuracy in Pavlovian-congruent conditions ("go to win" and "no-go to avoid losing"). No significant difference in Pavlovian bias was observed between the GNG ($M = 0.32$, $SD = 0.47$) and WMGNG ($M = 0.32$, $SD = 0.40$) tasks (paired t-test, $t(48) = -0.02$, $p = 0.986$, $d = 0.00$). However, participants could have been slower in learning the Pavlovian cue-outcome associations under WM load, as the instrumental learning became slower under WM load. Thus, it should be examined if participants expressed similar levels of Pavlovian bias in the GNG and WMGNG tasks after they learned the cue-outcome associations in both tasks. To this end, we plotted the temporal development of Pavlovian bias across trials (**S2 Fig**). We observed a delayed peak in the WMGNG compared to the GNG

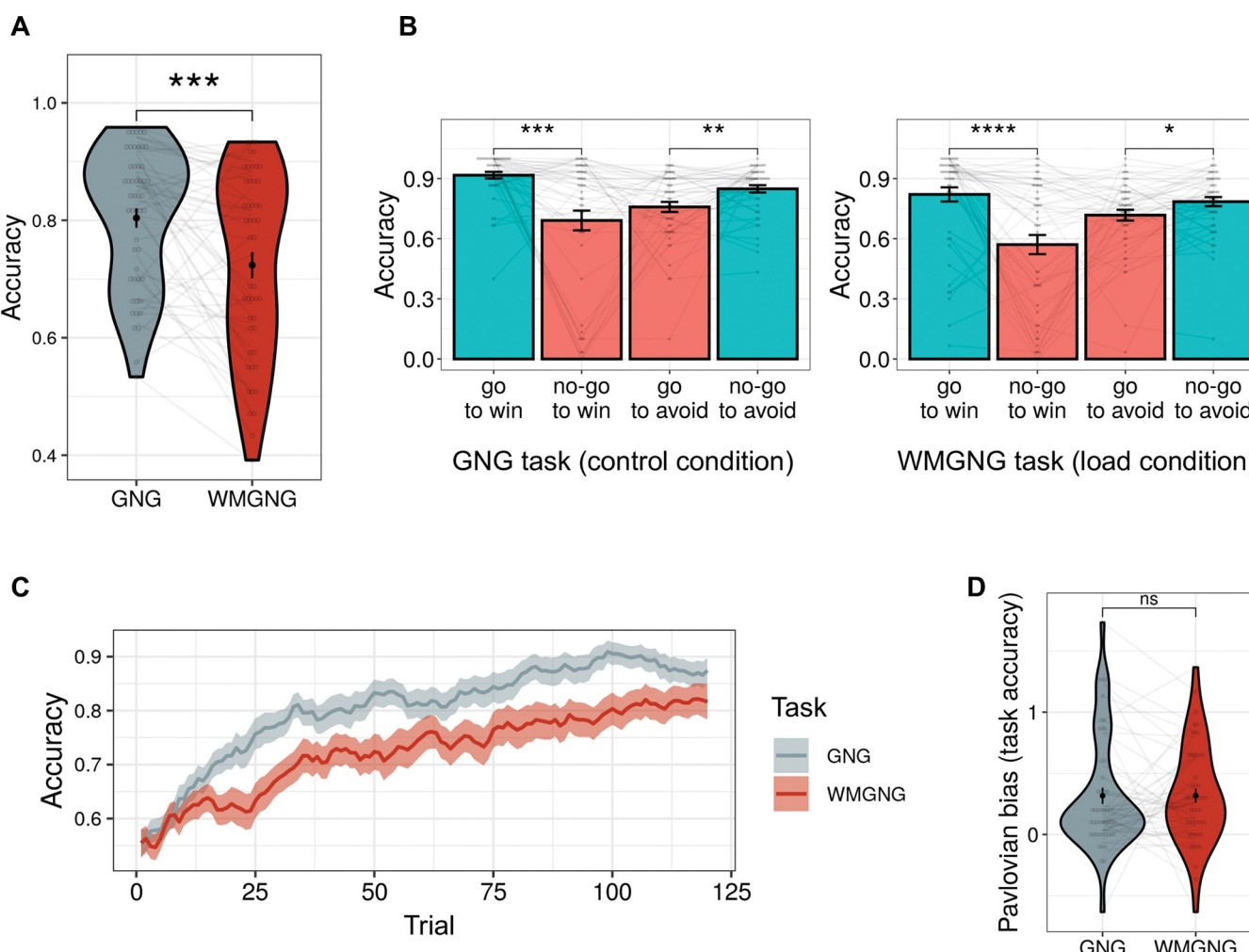

**Fig 2. Task performance (N = 49).** (A) Task accuracies (mean percentages of correct responses) in the GNG and WMGNG tasks show that participants performed better in the GNG task than in the WMGNG task. (B) Accuracy in each of the four trial types between the two tasks demonstrated that participants performed better in "go to win" and "no-go to avoid losing" trials (Pavlovian-congruent, blue) than in "no-go to win" and "go to avoid losing" trials (Pavlovian-incongruent, red). (C) The learning curve (i.e., the increase in accuracy across trials) was shallower in the WMGNG task than in the GNG task. Note that moving average smoothing was applied with filter size 5 to remove the fine variation between time steps. Lines indicate group means and ribbons indicate means ± standard errors of the means. (D) Pavlovian bias was calculated by subtracting accuracy in Pavlovian-incongruent conditions ("no-go to win" + "go to avoid losing") from accuracy in Pavlovian-congruent conditions ("go to win" + "no-go to avoid losing"). No significant difference in Pavlovian bias was observed between the GNG and WMGNG tasks. (A)-(B), (D) Black dots indicate group means and error bars indicate means ± standard errors of the means. Gray dots indicate individual accuracies; lines connect a single participant's performances. Asterisks indicate the results of pairwise t-tests. **** $p < 0.0001$, *** $p < 0.001$, ** $p < 0.01$, * $p < 0.05$.

task, which indicates that acquiring Pavlovian associations could have taken more time under WM load. Nonetheless, we observed similar levels of Pavlovian bias in both tasks after the initial peak. Thus, we concluded that our data do not show noticeable changes in Pavlovian bias under WM load.

## Computational modeling: WM load influences learning rate and irreducible noise

We used a computational modeling approach to test the three hypotheses. For this purpose, we developed eight nested models that assumed different learning rate, Pavlovian bias, or

irreducible noise parameters under WM load. These models were fitted to the data using hierarchical Bayesian analysis, then compared using the leave-one-out information criterion (LOOIC), where a lower LOOIC value indicates better out-of-sample predictive accuracy (i.e., better fit) [63]. Importantly, the use of computational modeling allowed us to test our hypothesis that WM load would increase random choices; this would have not been possible if we had performed behavioral analysis alone.

Based on earlier studies [45,64], we constructed a baseline model (model 1) that used a Rescorla-Wagner updating rule and contained learning rate ($\varepsilon$), Pavlovian bias, irreducible noise, go bias, and separate parameters for sensitivity to rewards and punishments (Materials and Methods). In the model, state-action values are updated with the prediction error; learning rate ($\varepsilon$) modulates the impact of the prediction error. Reward/punishment sensitivity ($\rho$) scales the effective size of outcome values. Go bias (b) and cue values weighted by Pavlovian bias ($\pi$) are added to the value of go choices. Here, as the Pavlovian bias parameter increases, the go tendency increases under the reward condition whereas the go tendency is reduced under the punishment condition; this results in an increased no-go tendency. Computed action weights are used to estimate action probabilities, and irreducible noise ($\xi$) determines the extent to which information about action weights is utilized to make decisions. As irreducible noise increases, action probabilities will be less reflective of action weights, indicating that action selection will become more random.

In models 2, 3, and 4, we assumed that WM load affects only one parameter. For example, in model 2, a separate Pavlovian bias parameter ($\pi_{wm}$) was assumed for the WM load condition. Models 3 and 4 assumed different learning rates ($\varepsilon_{wm}$) and irreducible noise ($\xi_{wm}$) parameters in their respective WM load conditions. In models 5, 6, and 7, we assumed that WM load would affect two parameters: model 5 had different Pavlovian bias ($\pi_{wm}$) and learning rate ($\varepsilon_{wm}$); model 6 had different Pavlovian bias ($\pi_{wm}$) and irreducible noise ($\xi_{wm}$); and model 7 had different learning rate ($\varepsilon_{wm}$) and irreducible noise ($\xi_{wm}$). Finally, model 8 was the full model, in which all three parameters were assumed to be affected by WM load.

The full model (model 8) was the best model (**Fig 3A** and **S2 Table**). In other words, participant behavior could be best explained when separate parameters were included for Pavlovian bias, learning rate, and irreducible noise parameters. Next, we analyzed the parameter estimates of the best-fitting model; we focused on comparing the posterior distributions of the parameters that were separately fitted in the two tasks (**Fig 3B**). The parameters were considered credibly different from each other if the 95% highest density intervals (HDI) of the two distributions showed no overlap [65]. **Fig 3B** illustrates that Pavlovian bias was not credibly different between the two tasks, consistent with our behavioral results that failed to show a change in Pavlovian bias under WM load. Conversely, the learning rate was credibly lower, while irreducible noise was credibly greater in the WMGNG than in the GNG task. These results support our hypotheses that WM load would reduce learning rate and that it would increase random choices. While the best model was the full model that assumed separate Pavlovian bias in the two tasks, no credible group difference was observed between these parameters. This is presumably because the full model was able to capture individual variations among participants (**S5 Fig**), despite the lack of credible difference in the group-level estimates between the two tasks. As expected, the 95% HDIs of go bias, reward sensitivity, and punishment sensitivity did not include zero, indicating that the participants exhibited go bias and reward/punishment sensitivity (see Supporting Information for the posterior distributions of individual parameters; **S4–S7 Figs**).

To further compare choice randomness between the two tasks, we examined the extent to which choices were dependent on value discrepancies between the two options. We first plotted the percentage of go choices for the GNG and WMGNG tasks by varying the quantiles of

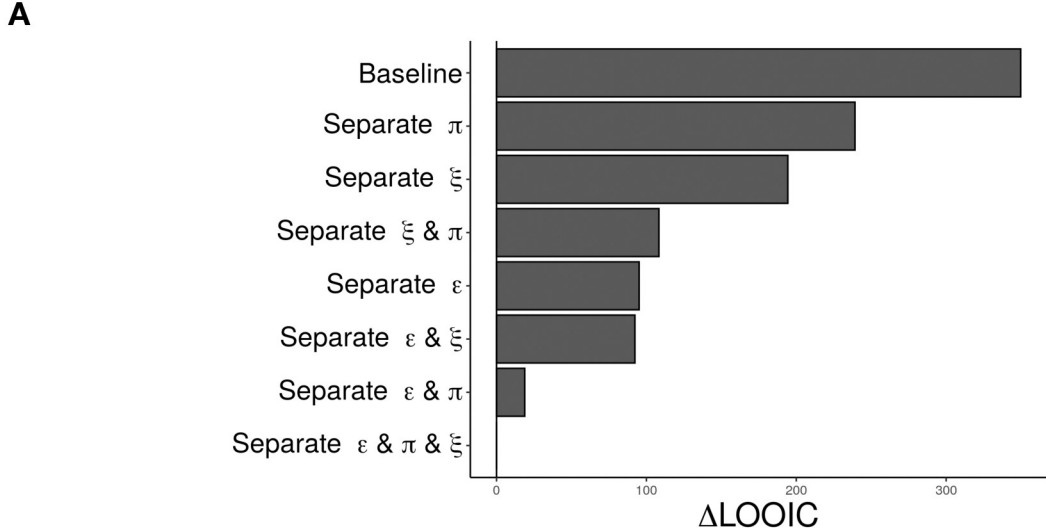

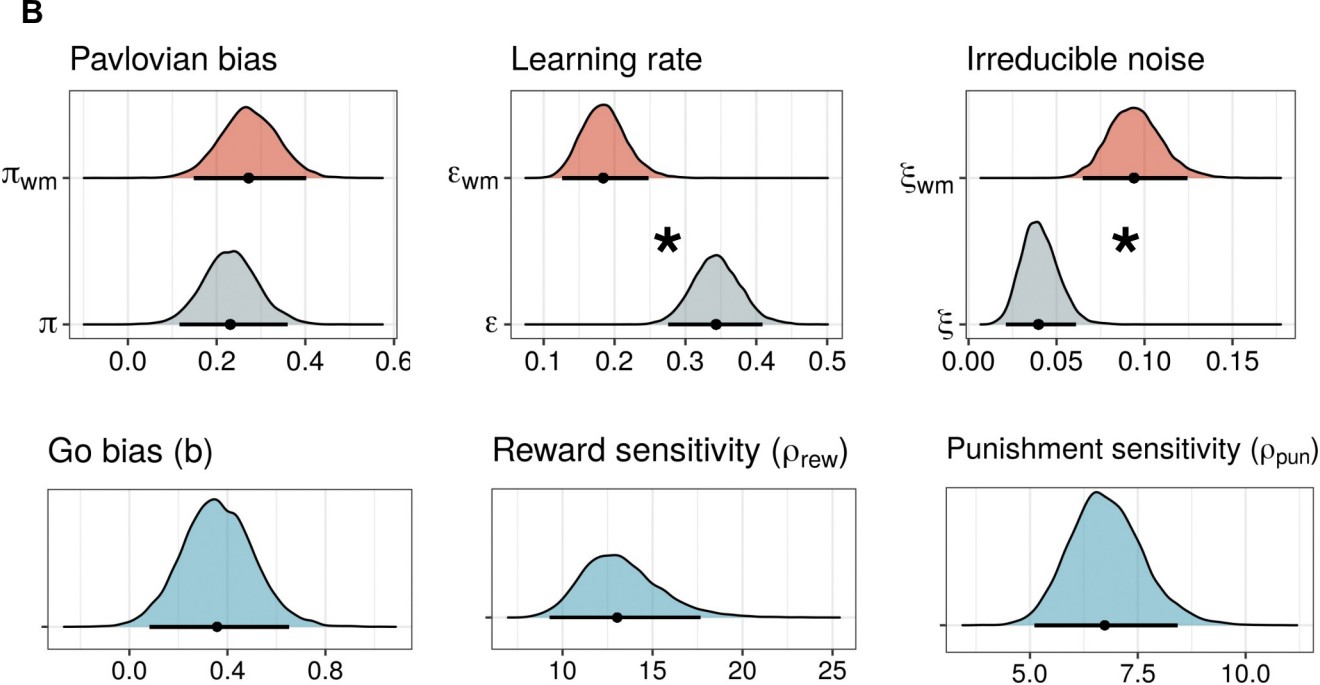

**Fig 3. Model comparison results and posterior distribution of the group-level parameters of the best-fitting model (N = 49).** (A) Relative LOOIC difference indicates the difference in LOOIC between the best-fitting model and each of the other models. The best-fitting model was the full model, which assumed separate Pavlovian bias, learning rate, and irreducible noise in GNG and WMGNG tasks. Lower LOOIC indicates better model fit. (B) Posterior distributions of group-level parameters from the best-fitting model. Learning rate and irreducible noise estimates were credibly different in the GNG and WMGNG tasks, while Pavlovian bias estimates were not. Dots indicate medians and bars indicate 95% HDIs. Asterisks indicate that the 95% HDIs of the two parameters' posterior distributions do not overlap (i.e., differences are credible).

differences in action weight between the "go" and "no-go" actions ($W_{go}$—$W_{nogo}$) (**Fig 4A**). The trial-by-trial action weights were extracted from the best-fitting model. Higher quantiles corresponded to a greater "go" action weight than "no-go" action weight. Overall, the go ratio increased from the first to the tenth quantile, indicating that the value differences between the

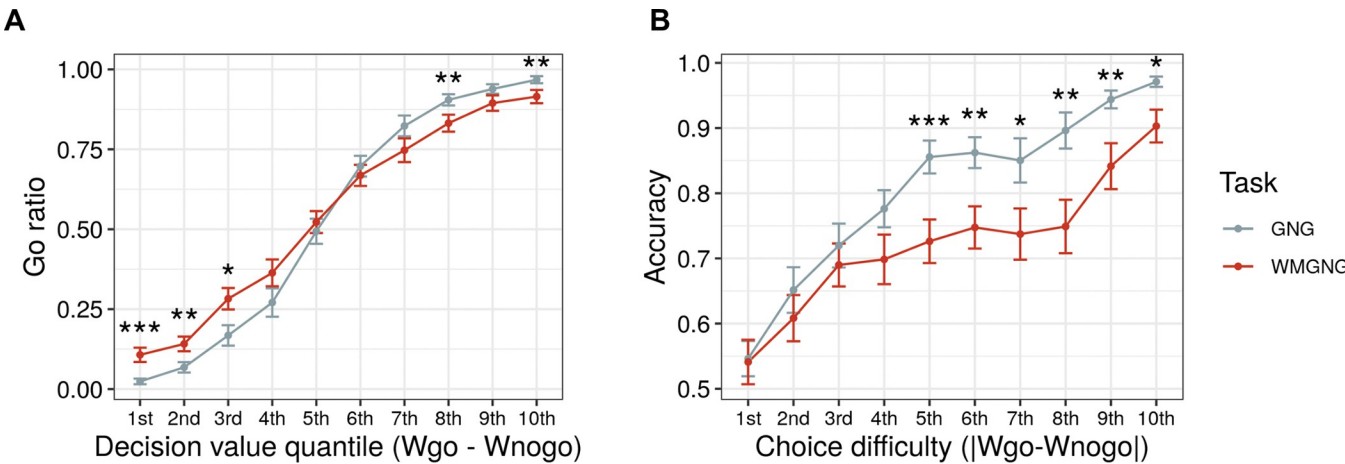

**Fig 4. Choice randomness (N = 49).** (A) Mean percentage of go choices for different quantiles of action weight differences ($W_{go}$—$W_{nogo}$) between "go" and "no-go" choices, where higher quantiles indicate higher decision values for "go" choices. Under WM load, the increase in go ratio according to quantile was less steep. (B) Mean accuracies for different quantiles of absolute value differences ($|W_{go}$—$W_{nogo}|$), where higher quantiles indicate larger value differences between two options or easier choices. Under WM load, the increase in accuracy according to quantile was less steep. (A)-(B) Dots are group means, and error bars are means ± standard errors of the means. Asterisks show the results of pairwise t-tests. **** $p < 0.0001$, *** $p < 0.001$, ** $p < 0.01$, * $p < 0.05$.

"go" and "no-go" actions affected participants' choices. This result further illustrates the difference between the two tasks: the increase in the go ratio was steeper in the GNG task than in the WMGNG task. In particular, the go ratio significantly differed between the two tasks for the first ($t(48) = -3.59$, $p = 0.001$, $d = 0.51$), second ($t(48) = -3.23$, $p = 0.002$, $d = 0.46$), third ($t(48) = -2.55$, $p = 0.014$, $d = 0.36$), eighth ($t(48) = 2.95$, $p = 0.005$, $d = 0.42$), and tenth ($t(48) = 2.76$, $p = 0.008$, $d = 0.39$) quantiles. Thus, under WM load, participants were less sensitive to the significant value difference between "go" and "no-go".

To compare these patterns in a different way and further explore the extent to which performance was dependent on choice difficulty, we plotted accuracies for the two tasks and for different quantiles of the absolute value differences ($|W_{go}$—$W_{nogo}|$; **Fig 4B**). We assumed that the choices would become easier when the absolute value difference was increased because a small value difference makes it difficult to choose between options. Overall, the accuracy increased from the first to the tenth quantile, indicating that participants performed better as the choices became easier. This result further illustrates the difference between the two tasks: the increase in accuracy was steeper in the GNG task than in the WMGNG task. Specifically, the accuracy significantly differed between the two tasks for the fifth ($t(48) = 4.12$, $p < 0.001$, $d = 0.59$), sixth ($t(48) = 2.95$, $p = 0.005$, $d = 0.42$), seventh ($t(48) = 2.44$, $p = 0.018$, $d = 0.35$), eighth ($t(48) = 3.13$, $p = 0.003$, $d = 0.45$), ninth ($t(48) = 2.87$, $p = 0.006$, $d = 0.41$), and tenth ($t(48) = 2.55$, $p = 0.014$, $d = 0.36$) quantiles. Thus, participants performed worse in the WM load condition than in the control condition when choices were easier. Overall, **Fig 4** demonstrates that WM load reduced the effect of the value difference on participants, indicating increased choice randomness.

Next, we examined if our model predicts the observed decrease in task performance under WM load both in the Pavlovian-congruent and Pavlovian-incongruent conditions. **S1B Fig** shows that our model indeed predicts the lower task performance in both types of conditions. This result is in line with our result that choice randomness increased under WM load. We can expect that an increase in randomness would result in a lower accuracy unless the accuracy was below the chance level in the first place. While the accuracy decrease could also be associated with the lower learning rate under WM load, we need to be cautious in this interpretation

because lower learning rates might instead increase the accuracy by making learning more robust against noise.

## Larger RPE signals in the striatum and weakened connectivity with prefrontal regions under WM load

Behavioral analysis revealed that WM load caused learning to occur more slowly but did not affect Pavlovian bias. The computational approach confirmed that the learning rate decreased; Pavlovian bias did not change under the load; and WM load led to increased choice randomness. Here, we sought to investigate the underlying neural correlates of these effects of WM load on learning rate, Pavlovian bias, and random action selection. First, we hypothesized that RPE signaling in the striatum would increase under WM load [15,18]. We conducted a model-based fMRI analysis using RPE as a regressor derived from the best-fitting model (see Materials and Methods for the full general linear models (GLMs) and regressor specifications). The RPE signal in the a priori striatum region of interest (ROI) was significantly greater in the WMGNG task than in the GNG task (contrast: RPE in WMGNG > RPE in GNG, MNI space coordinates $x = 13$, $y = 14$, $z = -3$, $Z = 3.96$, $p < 0.05$ small-volume corrected (SVC), **Fig 5A** and **S4 Table**). This supports our hypothesis that WM load would increase the contribution of the RL system and decrease the contribution of the WM system. We also tested the hypothesis that WM load would lead to a greater brain activation associated with Pavlovian bias, but we found no main effect of Pavlovian bias between the GNG and WMGNG tasks (WMGNG > GNG [Pavlovian-congruent > Pavlovian-incongruent]) within the striatum or SN/VTA ($p < 0.05$ SVC, a priori ROIs). Note that previous studies showed no significant result for the same contrast (Pavlovian-congruent > Pavlovian-incongruent) within the same regions [45]. As a test of our hypothesis that WM load would increase choice randomness, we examined whether the WM load led to less brain activation associated with value comparison (which would indicate a decreased contribution of the value comparison signal and thus relatively more contribution of randomness). We observed no main effect of WM load on random choice (WMGNG > GNG [W$_{chosen}$—W$_{unchosen}$]) within the ventromedial prefrontal cortex (vmPFC; $p < 0.05$ SVC, a priori ROI). See Supporting Information for further details regarding these findings (**S5 Table**).

Increased RPE signals under WM load may indicate reduced WM contribution and increased RL contribution to learning because of the load, suggesting diminished cooperation between the two systems for learning. Therefore, we conducted a psychophysiological interaction (PPI) analysis [66] using the gPPI toolbox [67] to test whether functional connectivity between areas associated with RL and WM systems would weaken under WM load. Specifically, we explored differences between the two tasks in terms of functional coupling between the striatum, which showed increased RPE signaling under WM load during the feedback phase, and other regions when computing reward expectations. The striatum showed significantly decreased connectivity with the vmPFC (MNI space coordinates $x = 13$, $y = 56$, $z = 0$, $Z = −4.90$, $p < 0.05$ whole-brain cluster-level familywise error rate (FWE)) and dlPFC (MNI space coordinates $x = -22$, $y = 63$, $z = 23$, $Z = −4.24$, $p < 0.05$ whole-brain cluster-level FWE, **Fig 5B** and **S6 Table**) in the WMGNG task, compared with the GNG task.

## Discussion

In this study, our main objective was to elucidate the neurocognitive effects of WM load on instrumental learning that involves Pavlovian–instrumental conflicts. We hypothesized that under WM load, 1) learning rate would decrease and RPE signals would become stronger, 2) Pavlovian bias would increase, and 3) action selection would become more random. First, we found that the limitation of WM resources with WM load led to a decrease in the learning rate

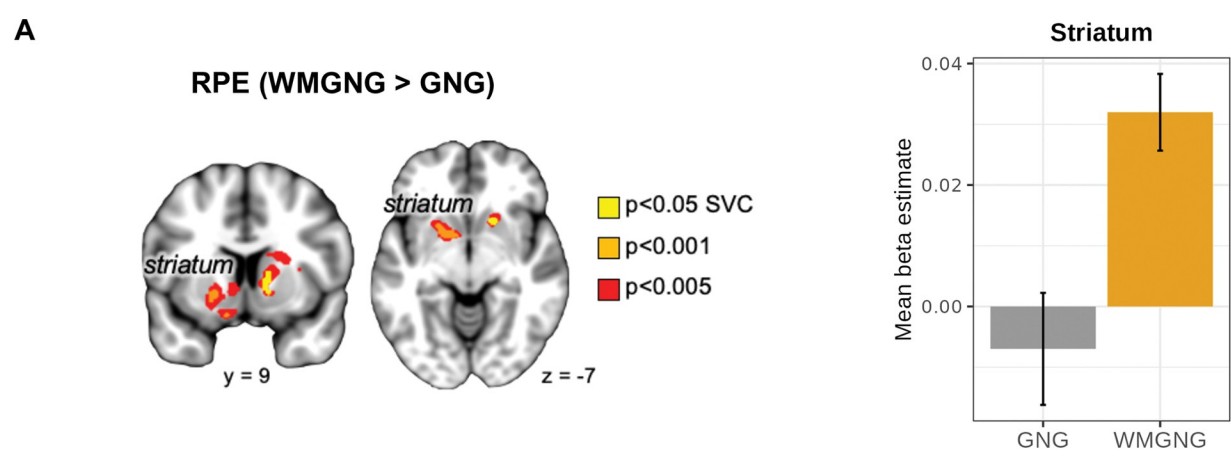

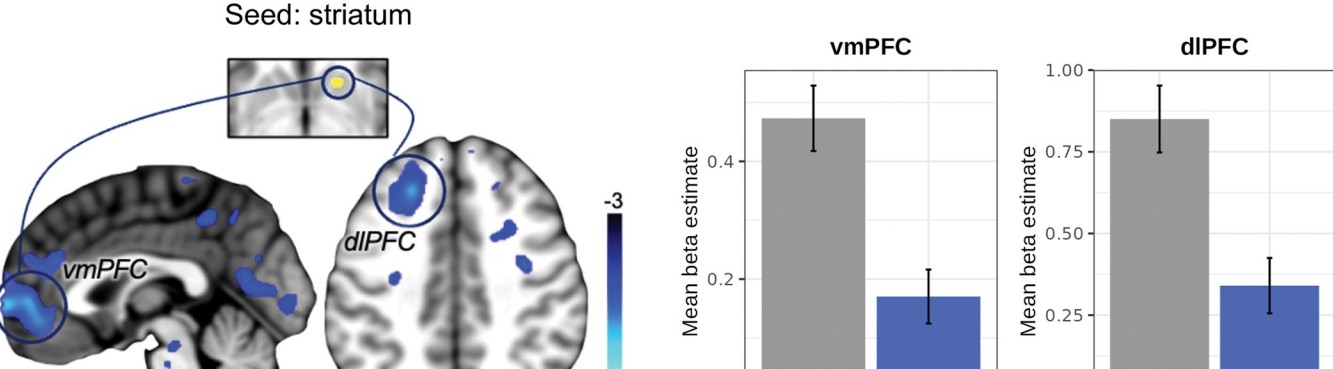

**Fig 5. fMRI results (N = 44).** (A) In the predefined ROI-based analysis, RPE signaling in the striatum was stronger in the WMGNG task than in the GNG task. The left figure shows significant regions at p < 0.05 (SVC) in yellow. The right figure shows an increased mean beta estimate in the striatum under WM load. (B) The left figure shows that functional connectivity between the striatum (seed region, top) and prefrontal regions, including vmPFC (bottom left) and dlPFC (bottom right), was weaker in the WMGNG task than in the GNG task when computing reward expectation ($p < 0.05$, whole-brain cluster-level FWE). The right figures show decreased mean beta estimates in vmPFC and dlPFC under WM load. In all figures, error bars are means ± standard errors of the means. Overlays are shown with a threshold of $p < 0.001$ (uncorrected). Color scale indicates t-values.

and increases in striatal RPE signals. The striatum, which subsequently showed stronger RPE signals under WM load, demonstrated weakened functional connectivity with prefrontal regions including the dlPFC and vmPFC, during reward prediction. WM load also increased random action selection. However, Pavlovian bias did not increase under WM load, suggesting that WM load did not affect the balance between Pavlovian and instrumental systems.

## Decreased contribution of the WM system and increased contribution of the RL system under WM load

The effect of WM load on instrumental learning remained consistent despite Pavlovian bias. In particular, our behavioral analysis revealed a shallower learning curve under WM load

(**Fig 2C**); modeling analysis confirmed that WM load reduced learning rate (**Fig 3**). We also found that RPE signaling in the striatum was strengthened under WM load (**Fig 5A**), consistent with previous findings [15,18].

These findings support theoretical explanations that WM load reduces the contribution of the WM system and increases the contribution of the RL system. In one such explanation, the rapid and immediate WM system learns in parallel with the slow and incremental RL system by directly storing the observed associations between states and actions [1,15–17,68]. WM and RL systems compete with each other based on their reliability in a given situation. Under WM load, the fast and capacity-limited WM system becomes less reliable; thus, the slow and incremental RL system supersedes the WM system, causing learning to occur more slowly and incrementally [1,16,30]. Such a shift toward slow, incremental learning can be adaptive because it is more reliable under resource-limited conditions and robust against noise in stationary environments [69]. Another explanation posits that RL computations are intertwined with WM; WM feeds reward expectations to the RL system [70–73] to improve reward prediction precision, thereby reducing RPEs and improving learning efficiency [15,18]. WM load constrains the WM system, which leads to a lower precision in reward prediction and thus greater RPEs.

These ideas are further supported by our finding that the striatum showed weakened functional connectivity with the dlPFC during reward prediction under WM load (**Fig 5A**). This result is in line with theoretical predictions (1) that the WM system would become unreliable and thus its contribution to instrumental learning would decrease and (2) that that the reward prediction information delivered from the WM to the RL system would become less precise and the RL system would rely less on it. However, further research is necessary to demonstrate the directionality of functional connectivity between the two systems during reward prediction; frontostriatal connectivity is reportedly bidirectional, such that the striatum may also provide prefrontal regions with inputs that relate to reward information [74,75].

Notably, we observed weakened connectivity between the vmPFC and the striatum. The vmPFC has been identified as a critical neural correlate of value-based decision-making; it integrates reward predictions [71], represents value signals or decision value [76–79], and affects reward anticipation/processing in the striatum [80,81]. Future research is needed to clarify which function of vmPFC was influenced by our WM load manipulation.

## No effect of WM load on Pavlovian bias

Contrary to our hypothesis, WM load did not influence Pavlovian bias. Behavioral and modeling results showed that Pavlovian bias did not significantly differ between the GNG and WMGNG tasks (**Figs 2D, 3** and **S2**), while fMRI analysis revealed that neural signaling associated with Pavlovian bias did not significantly differ between the two tasks (**S6 Table**). These findings indicate that the brain did not exhibit greater reliance on the computationally efficient system under WM load, in contrast to the results of previous studies [32]. We identified two possible explanations for this discrepancy. First, instrumental and Pavlovian learning in our task require similar amounts of WM resources; second, the WM system may not be related to modulating the balance between Pavlovian and instrumental systems.

In the first potential explanation, the amounts of WM resources necessary might have been similar for Pavlovian and instrumental learning in our task. Otto et al. showed that WM resource constraints promote the use of model-free compared to model-based system [32]. In their task, the model-based system required WM resources to retain the internal model of the task, which was its transition structure [20,82–85], whereas the model-free system did not maintain any model of the task [82,83,86]. Thus, WM resource requirement were quite

different between the two systems. In our paradigm, on the other hand, the WM resource requirements might not have been similar between the Pavlovian and the instrumental system. The Pavlovian system learned state-outcome associations, while the instrumental system learned state-action-outcome associations [42,87]. The difference in WM resource requirement was only the extra one dimension for the instrumental system. Instead of relying more on the Pavlovian system, which might only provide little boost in resource efficiency in our task, the participants may simply have favored slow, incremental learning, which requires small cognitive resources.

In the second potential explanation, WM resources may be unimportant with respect to modulating the Pavlovian–instrumental interaction, despite earlier studies' suggestions to the contrary. Several studies have proposed that prefrontal WM control systems are crucial for controlling Pavlovian bias. Electroencephalogram studies demonstrated that midfrontal theta oscillations are important for controlling Pavlovian bias [64,88], suggesting top-down prefrontal control over Pavlovian bias [64]. Furthermore, recruitment of the inferior frontal gyrus (IFG) is involved in appropriate response inhibition, helping to overcome Pavlovian bias [45]. Finally, there is indirect evidence that administration of levodopa, which increases dopamine levels, reduced Pavlovian influences on instrumental learning; such a reduction was speculated to result from increased dopamine levels in the PFC, which may have facilitated the operation of prefrontal WM functions [59]. A related finding suggested that genetic determinants of prefrontal dopamine function may be important in overcoming Pavlovian bias [62].

While the results of the present study appear to contradict these findings, several complex possibilities exist. In particular, although previous findings implied the involvement of prefrontal mechanisms (e.g., model-based prefrontal control [64] and WM [46,59] in controlling the Pavlovian system, they did not directly suggest active recruitment of the prefrontal WM system. First, while Cavanagh et al. speculated that midfrontal theta power could be indicative of "model-based top-down prefrontal control" [64], a subsequent study by Swart et al. suggested that midfrontal theta signals could only be involved in the detection of conflict by signaling "the need for control" [88,89], rather than being a source of direct control. Next, the IFG showed an increased BOLD response only in the "no-go" condition [45], implying that the IFG is important for "inhibitory" motor control (i.e., as a brake [90]); it does not participate in active maintenance or representation of goal-directed behaviors including both "go" and "no-go," which would be more closely associated with WM [29,91,92]. Finally, elevated dopamine levels should be cautiously interpreted as improvements in prefrontal WM function [59]. While dopamine has been shown to enable successful cognitive control in the prefrontal cortex, it may have three roles: gating behaviorally relevant sensory signals; maintaining and manipulating information in WM to guide goal-directed behavior; and relaying motor information to premotor areas for action preparation [93]. Moreover, distinct mechanisms have been known to modulate the influence of dopamine on WM in the PFC through distinct types of dopamine receptors [93]. Thus, there may be several ways to interpret the observation that dopamine level [59] or function [62] was associated with the modulation of Pavlovian influences. Considerable research is needed to fully understand the mechanisms by which dopamine levels affect Pavlovian bias. Alternatively, the role of prefrontal WM in controlling Pavlovian bias may not require vast resources. It may only be responsible for signaling a need for control [88], promoting response inhibition [45], or influencing subcortical areas (e.g., the striatum and subthalamic nucleus [94,95]).

Our result help refine the understanding of elevated Pavlovian bias in individuals suffering from certain mental disorders [48,49,96]. Researchers often assume the increased Pavlovian bias in clinical populations is, at least in part, a consequence of their impaired executive control, which hinders their goal-directed behavior [47,51]. However, the causal relationship

between executive control and the Pavlovian influence on behavior has not been actively investigated. In this study, we showed that constraining executive control with WM load did not increase the Pavlovian bias. It suggests that executive control deficits observed in clinical populations may not directly contribute to their heightened Pavlovian bias. Instead, clinical populations might perceive reward or punishment (or both) as more motivationally salient than healthy individuals, which leads to stronger urges for Pavlovian responses. Nonetheless, one should be cautious in this interpretation because the executive control impairments in clinical populations might be more severe than (or qualitatively different from) the ones elicited with WM load in our study.

One limitation of our study design is that the acquisition of Pavlovian cue-outcome associations and the expression of Pavlovian bias are not clearly separated. A more sophisticated way of comparing Pavlovian bias would first guarantee that participants acquired the cue-outcome associations to similar degrees in both tasks. Although we conducted a similar analysis (**S2 Fig**), it is not a complete solution to the issue because we cannot separate the two components with the data from our task. Future studies can address this issue by using a between-subject design, where the extra WM load are applied to only one group of participants after all participants perform enough trials of the GNG task. Another way would be to use the Pavlovian-instrumental transfer paradigm, which is similar to our paradigm but separates the Pavlovian learning, instrumental learning, and Pavlovian-instrumental transfer phases. Experimenters can manipulate the WM load during the transfer phase to examine the WM load effect on the expression of Pavlovian bias.

## Increased random choices under WM load

Another notable finding was that random choice increased under WM load. Our modeling analysis revealed that irreducible noise parameter estimates were greater in the WMGNG task than in the GNG task (**Fig 3**), suggesting increased random action selection under WM load. Further analysis using the modeling outputs revealed that participants' choices were less affected by the relative value difference between the "go" and "no-go" actions under WM load (**Fig 4A**). Moreover, analysis using the absolute difference between the two options (**Fig 4B**) revealed that the increase in accuracy became smaller as the absolute difference increased (i.e., the choice became easier). Both findings suggest that WM involvement led to an increase in random choices, regardless of value comparison and choice difficulty.

Our findings are broadly consistent with the results of previous studies concerning the role of WM and prefrontal regions in action selection and execution [97–104]. In particular, several studies have demonstrated that the interruption of WM function via WM load could increase the frequency of random choices in value-based decision-making tasks [34–36]. Additionally, transcranial direct current stimulation, a brain stimulation method, over the left PFC led to increased random action selection during an RL task, suggesting that the prefrontal WM component influenced action selection [105]. Furthermore, the importance of WM in action selection during learning tasks is supported by the indirect evidence that individual differences in WM capacity were correlated with appropriate exploratory action selection in multi-armed bandit tasks [106]. Overall, the reduced availability of WM resources because of WM load in our study may have compromised the participants' abilities to actively represent their current goals and actions, leading to reduced WM control over consistent choice based on value computation.

No significant neural correlates were identified with respect to the increased random choices. We assumed that random action selection would be associated with the reduced sensitivity to value difference or value comparison between the two options ("go" and "no-go")

[55]; thus, we hypothesized that value comparison signals would decrease under WM load. Contrary to our hypothesis, no significant differences in value comparison signaling in ROIs were observed between GNG and WMGNG tasks. There are several possible explanations for this null finding. Our assumption of value sensitivity may not be the source of the random choice observed here. Alternatively, subsequent attentional lapse [107,108] or value-independent noise [109] may have led to inconsistent action selection despite the presence of value comparison signals. Further research is necessary to distinguish these possibilities.

In summary, the present study has shown that WM load compromises overall learning by reducing learning speed via weakened cooperation between RL and WM; it also increases random action selection without affecting the balance between Pavlovian and instrumental systems. To our knowledge, this is the first study to investigate the neurocognitive effect of WM load during interactions between Pavlovian and instrumental systems. By investigating how learning and decision-making using different systems are altered in the presence of WM load and by linking such behaviors to their underlying neural mechanisms, this study contributes to our understanding of how distinct cognitive components interact with each other and synergistically contribute to learning. Because impairments in learning, balance among multiple systems, and action selection have been reported in various neurological and psychiatric disorders [2,110], our findings represent an important step toward improved understanding of various symptoms.

## Materials and methods

### Participants

Fifty-six adults participated in this study (34 women; 24.5±3.6 years old). All participants were healthy, right-handed; they had normal or corrected-to-normal visual acuity. They were screened prior to the experiment to exclude individuals with a history of neurological, or psychiatric illness. All participants provided written informed consent, and the study protocol was approved by the Institutional Review Board of Seoul National University.

The behavioral analysis included 49 participants (29 women; 24.3±3.3 y.o); the fMRI analysis included 44 participants (27 women; 24.2±3.3 y.o). Four participants were excluded because of technical issues; one participant was excluded because they slept during the task. Two participants were excluded because of poor performance in the 2-back task since the results in the dual-task paradigm could only be valid and interpretable when participants actually performed both tasks. The accuracy cutoff was 0.575, a value that rejects the null hypothesis that participants would randomly choose one of two options. After assessment of preprocessed image quality, five participants were excluded from the fMRI analysis because of head movements in the scanner, which can systematically alter brain signals; four out of these five were excluded because the mean framewise displacement exceeded 0.2 mm [111], while the remaining one was excluded after visual assessment of carpet plots [112].

### Experimental design and task

The experiment included two tasks: the original GNG task [45] and the GNG task paired with the 2-back task as a secondary task. The order of task completion was counterbalanced among participants; some participants performed the GNG task first while others began with the WMGNG task. The GNG and WMGNG tasks consisted of two blocks (four blocks in total); each block consisted of 60 trials. Therefore, each task contained 120 trials (240 trials in total). Participants underwent fMRI while performing the tasks for approximately 50 min, with a short (~60 s) break after each set of 60 trials. Before scanning, participants performed 10 practice trials for each of the GNG and WMGNG tasks to help them become accustomed to the

task structure and response timing. Participants received additional compensation based on their accuracy in the two tasks, along with the standard participation fee at the end of the experiment. We used different sets of fractal stimuli for the practice, first, and second tasks.

**Orthogonalized go/no-go (GNG) task.** Four trial types were implemented depending on the nature of the fractal cue (**Fig 1A**): press a button to gain a reward (go to win); press a button to avoid punishment (go to avoid losing); do not press a button to earn a reward (no-go to win); and do not press a button to avoid punishment (no-go to avoid losing). The meanings of fractal images were randomized among participants.

Each trial consisted of three phases: fractal cue presentation, response, and probabilistic outcome. **Fig 1B** illustrates the trial timeline. In each trial, participants were presented with one of four abstract fractal cues for 1000 ms. After a variable interval drawn from a uniform probability distribution within the range of 250–2000 ms, a white circle was displayed on the center of the screen for 1000 ms. When the circle appeared, participants were required to respond by pressing a button or not pressing a button. Next, the outcome was presented for 1000 ms: a green arrow pointing upwards on a ₩1000 bill indicated monetary reward, a red arrow pointing downwards on a ₩1000 bill indicated monetary punishment, and a yellow horizontal bar indicated no reward or punishment.

The outcome was probabilistic; thus, 80% correct responses and 20% incorrect responses resulted in the best outcome. Participants were instructed that the outcome would be probabilistic; for each fractal image, the correct response could be either "go" or "no-go," and they would have to learn the correct response for each cue through trial and error. The task included 30 trials for each of the four trial types (120 trials in total). Trial types were randomly shuffled throughout the duration of the task.

**Orthogonalized go/no-go + 2-back (WMGNG) task.** In the WM load condition, the GNG task was accompanied by a 2-back task to induce WM load. The combined task was named the WMGNG task; each trial had 2-back response and 2-back outcome phases after the GNG task (fractal cue, response, and probabilistic outcome). Participants were required to respond whether the cue in the current trial was identical to the cue presented in the two previous trials. For example, as shown in **Fig 1C**, the cue in the third trial differs from the cue in the first trial (two trials prior); thus, participants should respond "different" by pressing button after responding to the reinforcement learning task. In the fourth trial, they should respond "same." The positions of "SAME" and "DIFF" were randomized among participants.

## Computational modeling

**Baseline RL model with Pavlovian bias.** We adopted a previously implemented version of an RL model [45] that can model Pavlovian bias and choice randomness as well as learning rate. In our baseline model, we assumed no difference in parameters between the control and load conditions.

Expected values $Q(a_t, s_t)$ were calculated for each action a, "go" or "no-go", on each stimulus s (i.e., four trial types of the task) on each trial t. $Q(a_t, s_t)$ was determined by Rescorla-Wagner or delta rule updating:

$$Q_t(a_t, s_t) = Q_{t-1}(a_t, s_t) + \epsilon(\rho r_t - Q_{t-1}(a_t, s_t))$$

where ε is the learning rate. The learning rate (ε) is a step size of learning [4] that modulates how much of the prediction error, a teaching signal, is incorporated into the value update.

Rewards, neutral outcomes, and punishments were entered in the model through $r_t \in \{-1, 0, 1\}$, where ρ reflects the weighting (and effect sizes) of rewards and punishments. In all models, ρ could be different for rewards and punishments ($\rho_{rew}$ for gain, $\rho_{pun}$ for loss).

Action weights W($a_t$,$s_t$) were calculated from Q values, and the Pavlovian and go biases:

$$W_t(a_t, s_t) = \begin{cases} Q_t(a_t, s_t) + b + \pi V_t(s_t) & if\ a = go \\ Q_t(a_t, s_t), & else \end{cases}$$

where b was added to the value of go, while the expected value on the current state $V_t(s_t)$ was weighted by $\pi$ and added to the value of go choices. $V_t(s_t)$ was computed as follows:

$$V_t(s_t) = V_{t-1}(s_t) + \epsilon(\rho r_t - V_{t-1}(s_t)).$$

If the Pavlovian bias parameter ($\pi$) is positive, it increases the action weight of "go" in the reward conditions because $V_t(s_t)$ is positive. In the punishment conditions, positive $\pi$ decreases the action weight of "go" because $V_t(s_t)$ is negative. A value of $\pi$ close to 0 means that the agent relies mostly on the instrumental system (and the go bias), while a larger value of $\pi$ means that the Pavlovian system also has a great impact on the agent's action selection.

Action probabilities were dependent on these action weights W($a_t$,$s_t$), which were passed through a squashed softmax [4]:

$$P(a_t, s_t) = \left[ \frac{\exp[W(a_t, s_t)]}{\sum_{a'} \exp[W(a', s_t)]} \right] (1 - \xi) + \frac{\xi}{2}$$

where $\xi$ was the irreducible noise in the decision rule; it was free to vary between 0 and 1 for all models. The irreducible noise parameter explains the extent to which information about action weights is utilized in making a choice. As the irreducible noise increases, the influence of the difference between the action weights is reduced, indicating that action selection becomes random. We did not include the widely used inverse temperature parameter because $\rho_{rew}$ and $\rho_{pun}$ serve the same role as inverse temperature, deciding the level of determinism when generating actions based on action weights [56]. On the other hand, $\xi$ represents the constant randomness regardless of action weights, and has been widely used by researchers using similar models to ours [45,56,64,113,114].

**Additional models.** To test our hypotheses regarding the effects of WM load on parameters, we constructed seven additional nested models assuming different Pavlovian biases ($\pi$), learning rate ($\epsilon$), and irreducible noise ($\xi$) under WM load (**Table 1**). Model 1 is the baseline model. Model 2 assumed a separate Pavlovian bias parameter ($\pi$) for the WM load condition. Similarly, models 3 and 4 assumed different learning rates ($\epsilon$) and irreducible noises ($\xi$) in the WMGNG block, respectively. To address the possibility that two of the three parameters would be affected by the WM load, we constructed three additional models with eight free parameters: model 5 with different Pavlovian bias ($\pi$) and learning rate ($\epsilon$); model 6 with

**Table 1. Free parameters of all models.**

| Model No. | Model | # of parameters |
|:---:|:---|:---:|
| 1 | $\epsilon, \rho_{rew}, \rho_{pun}, b, \pi, \xi$ | 6 |
| 2 | $\epsilon, \rho_{rew}, \rho_{pun}, b, \pi, \xi, \pi_{wm}$ | 7 |
| 3 | $\epsilon, \rho_{rew}, \rho_{pun}, b, \pi, \xi, \epsilon_{wm}$ | 7 |
| 4 | $\epsilon, \rho_{rew}, \rho_{pun}, b, \pi, \xi, \xi_{wm}$ | 7 |
| 5 | $\epsilon, \rho_{rew}, \rho_{pun}, b, \pi, \xi, \pi_{wm}, \epsilon_{wm}$ | 8 |
| 6 | $\epsilon, \rho_{rew}, \rho_{pun}, b, \pi, \xi, \pi_{wm}, \xi_{wm}$ | 8 |
| 7 | $\epsilon, \rho_{rew}, \rho_{pun}, b, \pi, \xi, \epsilon_{wm}, \xi_{wm}$ | 8 |
| 8 | $\epsilon, \rho_{rew}, \rho_{pun}, b, \pi, \xi, \pi_{wm}, \epsilon_{wm}, \xi_{wm}$ | 9 |

different Pavlovian bias (π) and irreducible noise (ξ); and model 7 with different learning rate (ε) and irreducible noise (ξ). Finally, we constructed the full model, which assumed that all of these three parameters would be affected by WM load, leading to nine free parameters.

## Procedures for model fitting and model selection

Model parameters were estimated using hierarchical Bayesian analysis (HBA). Group-level distributions were assumed to be normally distributed, with mean and standard deviation parameters set as two free hyperparameters. We employed weakly informative priors to minimize the influences of those priors on the posterior distributions [65,115]. Additionally, for parameter estimation, the Matt trick was used to minimize the dependence between group-level mean and standard deviation parameters; it also facilitated the sampling process [116]. Moreover, bounded parameters such as learning rates and irreducible noise ($\in [0, 1]$) were estimated within an unconstrained space; they were then probit-transformed to the constrained space, thus maximizing MCMC efficiency within the parameter space [115,117].

We ran four independent chains with 4000 samples each, including 2000 warm-up samples (i.e., burn-in) to ensure that the parameters converged to the target distributions. Four chains were run to ensure that the posterior distributions were not dependent on initial starting points [118]. We visually checked convergence to target distributions by observing trace plots (**S3 Fig**) and computing the R statistics—a measure of convergence across chains [119]. R statistics were < 1.1 for all models, indicating that the estimated parameter values converged to their target posterior distributions (**S1 Table**).

Models were compared using the LOOIC, which is an information criterion calculated from the leave-one-out cross-validation [63]. This method is used to estimate the out-of-sample predictive accuracy of a fitted Bayesian model for model comparison and selection. The LOOIC is computed using the log-likelihood evaluated from posterior distributions or simulations of the parameters. The R package loo [63], which provides an interface for the approximation of leave-one-out cross-validated log-likelihood, was used to estimate the LOOIC for each model. Lower LOOIC values indicate better fit.

## fMRI scans: Acquisition and protocol

fMRI was performed on the same scanner (Simens Tim Trio 3 Tesla) using a 32-channel head coil across all participants. A high-resolution T1-weighted anatomical scan of the whole brain resolution was also acquired for each participant (TR = 2300ms, TE = 2.36ms, FOV = 256mm,1mm×1mm×1mm) to enable spatial localization and normalization. The participant's head was positioned with foam pads to limit head movement during acquisition. Functional data was acquired using echo-planar imaging (EPI) in four scanning sessions containing 64 slices (TR = 1500ms, TE = 30ms, FOV = 256mm, 2.3mm × 2.3mm × 2.3mm, multi-band acceleration factor = 4). For the GNG task, functional imaging data were acquired in two separate 277-volume runs, each lasting about 7.5 min. For the WMGNG task, data were acquired in two separate 357-volume runs, each lasting about 9.5 min.

## fMRI scans: General linear models

Preprocessing was performed using fMRIPrep 20.2.0 (RRID:SCR_016216) [120,121], which is based on Nipype 1.5.1 (RRID:SCR_002502) [122,123]. Details of preprocessing with fMRIPrep are provided in Supporting Information. Subsequently, images were smoothed using a 3D Gaussian kernel (8mm FWHM) to adjust for anatomical differences among participants. BOLD-signal image analysis was then performed using SPM12 [http://www.fil.ion.ucl.ac.uk/spm/] running on MATLAB v9.5.0.1067069(R2018b).

We built participant-specific GLMs, including all runs–two runs for the GNG block and two runs for the WMGNG block–and calculated contrasts to compare the two blocks at the individual level. The first-level model included six movement regressors to control the movement-related artifacts as nuisance regressors. Linear contrasts at each voxel were used to obtain participant-specific estimates for each effect. These estimates were entered into group-level analyses, with participants regarded as random effects, using a one-sample t-test against a contrast value of 0 at each voxel. The group-level model included covariates for gender, age, and the task order. For all GLM analyses, we conducted ROI analysis; the results were corrected for multiple comparisons using small volume correction (SVC) within ROIs.

*GLM1*: GLM1 was used to test the hypothesis that WM load would increase the contribution of the slow, incremental RL system because the resources for the fast WM system are constrained. We expected an stronger RPE signaling in the striatum in the WMGNG than the GNG task. GLM was implemented by the model-based fMRI approach and included the following regressors: (1) cue onset of "go to win" trials, (2) cue onset of "no-go to win" trials, (3) cue onset of "go to avoid losing" trials, (4) cue onset of "no-go to avoid losing" trials, (5) target onset of "go" trials, (6) target onset of "no-go" trials, (7) outcome onset, (8) outcome onset parametrically modulated by the trial-by-trial RPEs, and (9) wait onset (i.e., inter-trial interval). The regressor of interest was "RPE"; we compared the main effect of RPE between two tasks (RPE(8)WMGNG—RPE(8)$_{GNG}$). RPE regressors were calculated by subtracting the expected values (Q) from the outcome for each trial. Here, the outcome was the product of feedback multiplied by reward/punishment sensitivity. The a priori ROI was the striatum, which is widely known to process RPE [12,14,124]. The striatum ROI was constructed anatomically by combining the AAL3 definitions of bilateral caudate, putamen, olfactory cortex, and nucleus accumbens.

*GLM2*: GLM2 was used to test the hypothesis that neural responses associated with Pavlovian bias would increase under WM load. Specifically, the GLM examined whether the difference between the anticipatory response to fractal cues in Pavlovian-congruent trials and Pavlovian-incongruent trials was greater in the WMGNG task than in the GNG task in regions associated with Pavlovian bias. Therefore, GLM included the following regressors: (1) cue onset of "go to win" trials, (2) cue onset of "no-go to win" trials, (3) cue onset of "go to avoid losing" trials, (4) cue onset of "no-go to avoid losing" trials, (5) target onset of "go" trials, (6) target onset of "no-go" trials, (7) outcome onset of win trials, (8) outcome onset of neutral trials, (9) outcome onset of loss trials, (10) wait onset (i.e., inter-trial interval). We compared the main effect of Pavlovian bias (Pavlovian-congruent trials—Pavlovian-incongruent trials) between two tasks ([(1) + (4)—((2) + (3))]$_{WMGNG}$—[(1) + (4) ((2) + (3))]$_{GNG}$). The a priori ROIs were the striatum and SN/VTA, which are considered important in Pavlovian bias [45,46,52–54]. The striatum ROI was constructed anatomically by combining the AAL3 definitions of bilateral caudate, putamen, olfactory cortex, and nucleus accumbens. Furthermore, the SN/VTA was constructed by combining the AAL3 definitions of bilateral SN and VTA.

*GLM3*: GLM3 was used to the hypothesis that there would be less neural responses associated with value comparison signals under WM load. GLM3 was also implemented with a model-based fMRI approach: (1) cue onset of all trials, (2) cue onset parametrically modulated by the trial-by-trial decision values (W$_{chosen}$−W$_{unchosen}$), (3) target onset of "go" trials, (4) target onset of "no-go" trials, (5) outcome onset, and (6) wait onset (i.e., inter-trial interval). Decision value regressors were calculated by subtracting the action weights of the unchosen option (W$_{unchosen}$) from the action weights of the chosen option (W$_{chosen}$). We then compared the main effect of decision value between two blocks ((2)$_{WMGNG}$-(2)$_{GNG}$). The a priori ROI for GLM3 was the vmPFC, which was suggested as a region that encodes the relative chosen value (W$_{chosen}$−W$_{unchosen}$) [125,126]. Here, ROI masks were created by drawing a sphere with a

diameter of 10 mm around the peak voxel reported in the previous studies ([−6,48,−8] for vmPFC [125]).

*PPI analysis*: In addition to GLMs, we used PPI analysis to test whether WM load led to reduced cooperation between WM and RL systems for learning [15,18] by using PPI analysis. Here, to examine differences between the two blocks in terms of functional coupling between the prefrontal areas and the area computing RPE after choices, we performed PPI analysis using the gPPI toolbox [67]; the physiological variable was the time course of the striatum in the anticipation phase, and the psychological variable was the effect of WM load during the same phase. As a seed region (i.e., a physiological variable), the cluster striatum ROI (peak MNI space coordinates x = 13, y = 14, z = -3) was derived from the results of GLM1, which revealed stronger RPE signaling in the WMGNG task than in the GNG task during the feedback phase. The entire time series throughout the experiment was extracted from each participant in the striatum ROI. To create the PPI regressor, these normalized time series were multiplied by task condition vectors for the anticipation phase, which consisted of the cue representation and fixation phases as in GLM1. A GLM with PPI regressors of the seed region was thus generated together with movement regressors. The effects of PPI for each participant were estimated in the individual-level GLM; the parameter estimates represented the extent to which activity in each voxel was correlated with activity in the striatum during the anticipation phase. The contrast was constructed by subtracting activity during the anticipation phase in the GNG task from activity in the WMGNG task (WMGNG vs. GNG in the anticipation phase). Individual contrast images for functional connectivity were then computed and entered into one-sample t-tests in a group-level GLM together with nuisance covariates (i.e., gender, age, and task order). Whole-brain cluster-level FWE correction was applied for PPI analysis.

## Corrections for multiple comparisons

For each GLM, we performed whole-brain FWE corrections at the cluster level (corrected p < 0.05, with the height threshold of p < 0.001) for multiple comparisons [127,128] using SPM12. For a priori ROIs, we used SVC using SPM12.

## Supporting information

**S1 Text. fMRIPrep details.**
(DOCX)

**S2 Text. fMRI results details.**
(DOCX)

**S1 Fig. Performance in the Pavlovian-congruent and Pavlovian-incongruent conditions (N = 49).** (A) Participants' accuracy in the Pavlovian-congruent and Pavlovian-incongruent conditions, separately for the GNG and WMGNG tasks. Participants showed lower accuracy in the WMGNG compared to the GNG task both in the Pavlovian-congruent and Pavlovian-incongruent conditions. (B) Prediction from the best model regarding the accuracy in the Pavlovian-congruent and Pavlovian-incongruent conditions. We used a one-step ahead prediction for generating the model predictions. In line with our participants' data, our model predictions showed lower accuracy in the WMGNG than the GNG task in both types of conditions. In all figures, error bars indicate means ± standard errors of the means.
(PNG)

**S2 Fig. Temporal development of Pavlovian bias (N = 49).** The behavioral measure of Pavlovian bias, which is the difference in accuracy between the Pavlovian-congruent and

Pavlovian-incongruent conditions, plotted as a function of trial separately for the GNG and WMGNG tasks. While the peak in the WMGNG task appeared later than the GNG task, Pavlovian bias was similar between the tasks after the peaks. Moving average smoothing was applied with filter size 3. Lines indicate group means and ribbons indicate means ± standard errors of the means. Squares indicate the peak values of Pavlovian bias during the two tasks. (PNG)

**S3 Fig. Trace plots of group parameters for the best-fitting model.** The trace plots show that MCMC samples were well mixed and converged. Note that the plots excluded burn-in samples.
(PNG)

**S4 Fig. Posterior distributions of individual go bias, reward sensitivity, punishment sensitivity parameters.** Dots indicate medians and thick bars indicate 95% HDIs.
(PNG)

**S5 Fig. Posterior distributions of individual Pavlovian bias and Pavlovian bias under WM load parameters.** Dots indicate medians and thick bars indicate 95% HDIs.
(PNG)

**S6 Fig. Posterior distributions of individual learning rate and learning rate under WM load parameters.** Dots indicate medians and thick bars indicate 95% HDIs.
(PNG)

**S7 Fig. Posterior distributions of individual irreducible noise and irreducible noise under WM load parameters.** Dots indicate medians and thick bars indicate 95% HDIs.
(PNG)

**S8 Fig. Results of fMRI replication analyses.** (A) The main effect of WM load during the anticipation phase. As reported in previous meta-analysis studies [29,129], under WM load, regions of the lateral prefrontal cortex (PFC), including bilateral superior gyrus (MNI space coordinates x = -22, y = 0, z = 50, Z = 6.03, $p < 0.05$ whole-brain cluster-level family-wise error (FWE)) and middle frontal gyrus (MNI space coordinates x = 29, y = 2, z = 57, Z = 5.44, $p < 0.05$ whole-brain cluster-level FWE), and left precentral gyrus (MNI space coordinates x = -45, y = 5, z = 30, Z = 5.79, $p < 0.05$ whole-brain cluster-level FWE) and left inferior parietal cortex (MNI space coordinates x = -38, y = -51, z = 41, Z = 5.42, $p < 0.05$ whole-brain cluster-level FWE) showed increased BOLD signal. These results indicate that participants indeed had cognitive loads in the brain level. (B) The main effect of reward outcome. The reward was significantly associated with the signal in the striatum (MNI space coordinates x = 18, y = 5, z = -12, Z = 4.47, $p < 0.05$ whole-brain cluster-level FWE) and ventromedial PFC (MNI space coordinates x = -3, y = 68, z = 4, Z = 4.92, $p < 0.05$ whole-brain cluster-level FWE). (C) The main effect of loss outcome. Loss-related regions such as the insula (MNI space coordinates x = -36, y = 19, z = -10, Z = 6.77, $p < 0.05$ whole-brain cluster-level FWE) showed increased BOLD response. (B) and (C) are consistent with the previous findings suggesting reward- and loss-related regions [130,131]. Overlays are shown with a threshold of $p < 0.001$ (uncorrected). Color scale indicates t-values.
(PNG)

**S1 Table. Statistics of posterior distributions of group parameters for the best-fitting model.** $\hat{R}$ values [119] for all parameters were close to 1.00 ($< 1.1$), which indicates that the estimated parameter values converged to their target posterior distributions.
(XLSX)

**S2 Table. LOOIC for each model.**
(XLSX)

**S3 Table. Whole-brain results from replication analyses.** Whole-brain cluster-level family-wise error (FWE) for multiple comparison with a cluster-forming threshold of $p < 0.001$. * The insula (peak in insula: x = -36, y = 19, z = -10, $Z = 6.77$) reported in the result (**S6C Fig**) was included in this cluster of which the peak coordinate was located in orbital part of IFG.
(XLSX)

**S4 Table. ROI and whole-brain results of the main effect of WM load on RPE.** * p: small-volume corrected FWE within an anatomical striatum ROI defined from aal3 atlas with a cluster-forming threshold of $p < 0.001$. ** p: whole-brain cluster-level FWE for multiple comparison with a cluster-forming threshold of $p < 0.001$.
(XLSX)

**S5 Table. Neural statistics of the effect of WM load on Pavlovian bias.**
(XLSX)

**S6 Table. Whole-brain results from PPI analysis.** * The dlPFC (peak in dlPFC: x = -22 y = 63, z = 23, Z = -4.24) reported in the result (**Fig 5B**) was included in this cluster of which peak coordinate was located in medial SFG.
(XLSX)

## Author Contributions

**Conceptualization:** Heesun Park, Hoyoung Doh, Harhim Park, Woo-Young Ahn.

**Data curation:** Heesun Park, Hoyoung Doh, Eunhwi Lee, Harhim Park.

**Formal analysis:** Heesun Park, Hoyoung Doh, Eunhwi Lee.

**Funding acquisition:** Woo-Young Ahn.

**Investigation:** Heesun Park, Hoyoung Doh, Eunhwi Lee, Harhim Park.

**Methodology:** Heesun Park, Hoyoung Doh, Eunhwi Lee, Harhim Park, Woo-Young Ahn.

**Project administration:** Heesun Park, Woo-Young Ahn.

**Resources:** Woo-Young Ahn.

**Software:** Heesun Park, Hoyoung Doh, Eunhwi Lee, Harhim Park, Woo-Young Ahn.

**Supervision:** Woo-Young Ahn.

**Validation:** Heesun Park, Hoyoung Doh, Eunhwi Lee, Harhim Park.

**Visualization:** Heesun Park, Hoyoung Doh, Eunhwi Lee.

**Writing – original draft:** Heesun Park.

**Writing – review & editing:** Heesun Park, Hoyoung Doh, Eunhwi Lee, Woo-Young Ahn.

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
