## [Decision Letter · Decision Letter 0]

9 Jun 2023

Dear Doh,

Thank you very much for submitting your manuscript "The neurocognitive role of working memory load when Pavlovian motivational control affects instrumental learning" for consideration at PLOS Computational Biology.

As with all papers reviewed by the journal, your manuscript was reviewed by members of the editorial board and by several independent reviewers. In light of the reviews (below this email), we would like to invite the resubmission of a significantly-revised version that takes into account the reviewers' comments.

**Specifically, both reviewers raise important issues with regard to interpretation of the results and conclusion of the manuscript, both of which warrant additional analyses as suggested by the reviewers. **

We cannot make any decision about publication until we have seen the revised manuscript and your response to the reviewers' comments. Your revised manuscript is also likely to be sent to reviewers for further evaluation.

Sincerely,

Alireza Soltani

Academic Editor

PLOS Computational Biology

Thomas Serre

Section Editor

PLOS Computational Biology

Reviewer's Responses to Questions

**Comments to the Authors:**

Reviewer #1: In this well written and thorough study the authors investigated whether Pavlovian biases would increase under working memory load. It is generally accepted that 'Pavlovian biases' (the tendency to find it more difficult to make approach or avoid instrumental responses towards stimuli that signal punishment or reward respectively) are overcome by goal-directed control. Thus, it follows that working memory load should lead to an increase in those biases. However, the authors suggest using both behavioural and neuroimaging data, that this is not the case. Regardless of working memory load, the Pavlovian biases and their interference with ongoing instrumental behaviour remained static. I find the protocol well detailed and the analyses clear. I agree with the author's comments in the discussion that the evidence for the assumed model (of goal-directed control constraining Pavlovian biases) is to date fairly speculative and thus perhaps not valid. However, I am not entirely sure whether we can confidently accept the conclusions of the study as they currently stand.

- WMload would not only affect the learning of the correct instrumental response to make (go/no-go) but also learning about which outcome is signalled by the cue (potential reward or punishment). The rate of learning about the cue-outcome relationships directly affects the expression of Pavlovian biases and thus these biases could be expected to emerge more slowly in those who are under WM load compared to those who are not. However, this nuance is not captured in the current analyses where the order of tasks is counterbalanced and collapsed (because task order is not a parameter in the computational model). I think a stronger test of the hypotheses would be the comparison of Pavlovian biases at the end of task 1 vs. beginning of task 2 for the subset of participants who did the tasks in the order GNG then WMGNG. Does WM load lead to an immediate increase in Pavlovian bias? Another option would be to first ensure no difference in Pavlovian biases at the end of task 1 between the two groups. Even if the biases emerged more slowly in the WMGNG group, they may still be comparable at the end of the first task. Differences in performance could then be examined between the two groups during task 2. This would directly address the question of "once these Pavlovian biases have been learned, does WM load increase the expression of them"? I realise that the study was not powered for between group analyses, but if the authors wish to make the claim that WM load does not affect the expression of Pavlovian bias then the effects of WM load on cue-outcome learning vs. bias expression need to be separated out. It seems that a between groups design would have been better to test these hypotheses (where all participants are first given the opportunity to learn the relationships without WM load and then half of them are placed under WM load for the second half). A between-groups design was indeed used by Otto to examine the balance of model-free/model-based learning which the authors refer to repeatedly. The authors should discuss the current design as a limitation and perhaps tone down the interpretation of the current findings unless I have missed something critical?

- If we do believe that WM load does not affect the expression of Pavlovian biases then what does this mean for our clinical models? As the authors mention, numerous studies have demonstrated that the Pavlovian biases are stronger in those suffering from various clinical disorders. Deficits in executive control are often identified as the culprit here, but the results of this study (if true) would suggest instead that reward and punishment are more motivationally salient in these groups and that executive control deficits are largely irrelevant. This is a very interesting conclusion which I think could be explored a little more in the discussion.

Reviewer #2: Park and colleagues present a study investigating how working memory, instrumental learning, and pavlovian approach interact. They found that increased working memory load impairs instrumental learning, while not influencing pavlovian approach. Using fMRI, they found enhanced RL-related signals in the striatum and reduced PFC-striatal connectivity under WM load. They conclude that WM load impairs instrumental learning while not affecting Pavlovian approach. Overall, I am enthusiastic about the manuscript: the study is well-designed, and the analyses are sophisticated and appropriate. I do, however, have some issues with the interpretation of the results.

1) I found the term “Pavlovian influence” confusing because there are two pavlovian effects: the reward/go effect and the effects on task learning. The former is not really Pavlovian as much as it is a motivational congruency effect. I am not attached to specific terms here, it is just very confusing to track these separate ideas throughout the paper.

2) The authors refer to lower learning rate as “slower learning” several times. This is common in the field, but is not really accurate, and in a way that is important for this paper: lower learning rates integrate over longer time horizons than faster learning rates. This has a tradeoffs (i.e., noise has less of an influence, better memory, but slower to recognize shifts in reward structure). There are no shifts in reward structure here, so lower learning rates are primarily adaptive in the sense that they can compensate for the short memory capacity of WM. The authors should give more attention to the subtlety in this parameter. This is particularly relevant to the finding that behavioral learning rates were reduced whereas striatal correlates of learning were increased under WM load.

3) The above-two points combined lead to much confusion and make me disagree with some key assertions in the paper: For example, from the abstract, the authors say: “These results suggest that the limitation of cognitive resources by WM load decelerates instrumental learning through the weakened cooperation between WM and RL; such limitation also makes action selection more random, but it does not directly affect the balance between instrumental and Pavlovian systems.” What does it mean for instrumental learning to decelerate? The model does not test the balance between the systems, and that finding is based on the go-congruency effect (which is unrelated to instrumental learning, at least in their model). If anything, there is evidence for a change in the balance of the systems from the PPI results.

4) For the PPI results, the authors should plot the baseline connectivity values (perhaps from ROIs). If the baseline connectivity is negative, then the result would indicate increased negative connectivity, which would have a different interpretation.

5) The authors should be more cautious about reverse inference when referring to the striatal prediction error signal.

6) The main test is missing detail about the whole-brain and SVC correction procedures

7) Why is the olfactory bulb in the striatum ROI?

8) The authors use a different noise parameter than is common in the literature (softmax noise). I am curious for the authors to explain their rationale behind this choice.

9) Why would participants perform worse when choices are easier in WM condition? That seems counterintuitive. Does the model predict this behavior?

10) I believe (though I could be mistaken) the authors need to submit their raw data to OpenNeuro or some other repository as well as more public info on the neuroimaging code in order to be compliant with journal policies. The neurovault maps are not enough to be able to reproduce the analyses.

Minor

1) Keeping track of the hypotheses by number is confusing. It would be better to repeat the hypotheses

2) What was the smoothing kernel (this should be in the main text)

3) What was the multiband/acceleration protocol for the fMRI data?

4) Several times the authors use “whole-based” when I think they mean “whole-brain”

5) Is the b parameter a go bias?

**Have the authors made all data and (if applicable) computational code underlying the findings in their manuscript fully available?**

Reviewer #1: Yes

Reviewer #2: **No: **I believe (though I could be incorrect) the authors need to submit their raw fMRI data to OpenNeuro or some other repository as well as neuroimaging code in order to be compliant with journal policies. The neurovault maps are not enough to be able to reproduce the neuroimaging analyses. Behavioral data and code are publicly available as required.

PLOS authors have the option to publish the peer review history of their article (what does this mean?). If published, this will include your full peer review and any attached files.

Reviewer #1: No

Reviewer #2: No
---

## [Decision Letter · Decision Letter 1]

15 Nov 2023

Dear Prof. Ahn,

We are pleased to inform you that your manuscript 'The neurocognitive role of working memory load when Pavlovian motivational control affects instrumental learning' has been provisionally accepted for publication in PLOS Computational Biology.

Best regards,

Alireza Soltani

Academic Editor

PLOS Computational Biology

Thomas Serre

Section Editor

PLOS Computational Biology

Reviewer's Responses to Questions

**Comments to the Authors:**

Reviewer #1: Thank you, the work is interesting albeit not perfectly designed! But this is now clearer in the revised version.

Reviewer #2: The authors have satisfied my concerns. Congratulations on a great paper!

**Have the authors made all data and (if applicable) computational code underlying the findings in their manuscript fully available?**

Reviewer #1: None

Reviewer #2: Yes

PLOS authors have the option to publish the peer review history of their article (what does this mean?). If published, this will include your full peer review and any attached files.

Reviewer #1: No

Reviewer #2: No

---

## [Editor Report · Acceptance letter]

5 Dec 2023

PCOMPBIOL-D-23-00532R1 

The neurocognitive role of working memory load when Pavlovian motivational control affects instrumental learning

Dear Dr Ahn,

I am pleased to inform you that your manuscript has been formally accepted for publication in PLOS Computational Biology. Your manuscript is now with our production department and you will be notified of the publication date in due course.

With kind regards,

Zsofia Freund
